# Human Milk Exosomal MicroRNA: Associations with Maternal Overweight/Obesity and Infant Body Composition at 1 Month of Life

**DOI:** 10.3390/nu13041091

**Published:** 2021-03-27

**Authors:** Kruti B. Shah, Steven D. Chernausek, Lori D. Garman, Nathan P. Pezant, Jasmine F. Plows, Harmeet K. Kharoud, Ellen W. Demerath, David A. Fields

**Affiliations:** 1Section of Pediatric Diabetes and Endocrinology, Department of Pediatrics, University of Oklahoma Health Sciences Center, Oklahoma City, OK 73104, USA; steven-chernausek@ouhsc.edu (S.D.C.); david-fields@ouhsc.edu (D.A.F.); 2Harold Hamm Diabetes Center, University of Oklahoma Health Sciences Center, Oklahoma City, OK 73104, USA; 3Oklahoma Medical Research Foundation, Department of Genes and Human Disease, Oklahoma City, OK 73104, USA; Lori-Garman@omrf.org (L.D.G.); Nathan-Pezant@omrf.org (N.P.P.); 4Department of Endocrinology, The Saban Research Institute, Children’s Hospital Los Angeles, Los Angeles, CA 90027, USA; plows@usc.edu; 5Division of Epidemiology and Community Health, University of Minnesota, Minneapolis, MN 55454, USA; kharo001@umn.edu (H.K.K.); ewd@umn.edu (E.W.D.)

**Keywords:** breast milk microRNA, exosomes, maternal obesity, infant growth and body composition

## Abstract

Among all the body fluids, breast milk is one of the richest sources of microRNAs (miRNAs). MiRNAs packaged within the milk exosomes are bioavailable to breastfeeding infants. The role of miRNAs in determining infant growth and the impact of maternal overweight/obesity on human milk (HM) miRNAs is poorly understood. The objectives of this study were to examine the impact of maternal overweight/obesity on select miRNAs (miR-148a, miR-30b, miR-29a, miR-29b, miR-let-7a and miR-32) involved in adipogenesis and glucose metabolism and to examine the relationship of these miRNAs with measures of infant body composition in the first 6 months of life. Milk samples were collected from a cohort of 60 mothers (30 normal-weight [NW] and 30 overweight [OW]/obese [OB]) at 1-month and a subset of 48 of these at 3 months of lactation. Relative abundance of miRNA was determined using real-time PCR. The associations between the miRNAs of interest and infant weight and body composition at one, three, and six months were examined after adjusting for infant gestational age, birth weight, and sex. The abundance of miR-148a and miR-30b was lower by 30% and 42%, respectively, in the OW/OB group than in the NW group at 1 month. miR-148a was negatively associated with infant weight, fat mass, and fat free mass, while miR-30b was positively associated with infant weight, percent body fat, and fat mass at 1 month. Maternal obesity is negatively associated with the content of select miRNAs in human milk. An association of specific miRNAs with infant body composition was observed during the first month of life, suggesting a potential role in the infant’s adaptation to enteral nutrition.

## 1. Introduction

Breastfeeding is one of the most highly effective preventive measures that a mother can take to protect the health of her infant and herself (US Surgeon General, January 2011). The incidence of obesity and diabetes are rising, and multiple lines of evidence suggest that origins of obesity occur in the prenatal and early postnatal periods [1,2,3]. Exclusive and longer duration of breastfeeding plays a role in protecting against later obesity and type 2 diabetes mellitus, relative to formula [4,5,6]. There is also growing evidence that human milk (HM) is not only a source of macronutrients, but also of bioactive factors with the potential to influence the growth and body composition of the child [7,8]. Though bioactive factors are being discovered in HM, the mechanisms by which breast milk plays a role in reducing risks of obesity and type 2 diabetes mellitus is unknown. Our group and others have identified specific, non-nutritive substances in breast milk, the abundance of which are affected by maternal obesity and metabolic status [7,9,10]. As an outgrowth of this work, we have begun to explore a novel mechanism involving the transfer of miRNAs via HM from the mother to the developing infant.

Among all body fluids, HM is one of the richest sources of miRNAs [11]. These small, non-coding RNAs bind to complementary sequences within the 3′ untranslated region of messenger RNAs (mRNAs) and thereby modulate protein production, typically by degrading or repressing translation of targeted mRNAs [12,13,14,15]. They have been shown to function in multiple biological processes, including pathways involved in adipocyte differentiation, insulin signaling, and glucose metabolism [16,17]. miRNAs in HM can be isolated from skim milk, milk fat, and cells, or are found within small extracellular vesicles called exosomes [18]. Exosomal miRNAs in HM are protected from enzymatic degradation, [19] allowing uptake by the infant’s intestinal epithelial cells [20,21]. It has been shown that gene expression of intestinal epithelial cells is altered by exposure to milk exosomes in vitro [22]. Thus, ingested exosomal miRNAs may affect gene transcription in the infant gut to regulate both nutrient uptake, and growth and development of the infant. Furthermore, milk miRNAs may be absorbed by the immature intestinal tract of the infant and enter the circulation, where they may exert their action on a systemic level [23,24].

Given the possible functional role of these exosomal miRNAs in the growing infant, it remains crucial to understand maternal factors that might impact miRNA profile in breast milk. Limited studies have examined the impact of maternal factors such as maternal diet [25] and overweight/obesity [26,27] on miRNA profile in breast milk. Further, studies examining the role of exosomal miRNAs on early life infant growth measures are sparse [27]. In the present pilot study, our focus was on six miRNAs known to be involved in insulin signaling and adipogenesis pathways based on previous literature: miR-148a-3p, miR-30b-5p, miR-let-7a-5p, miR-29a-3p, miR-29b-3p, and miR-32-3p [16,28,29]. The objectives of the study were [1] test the hypothesis that maternal obesity is associated with altered abundance of specific HM exosomal miRNAs involved in insulin signaling, glucose homeostasis, and adipogenesis, and [2] examine relationships between HM miRNA and infant anthropometrics and body composition during the first six months of life.

## 2. Materials and Methods

### 2.1. Participants and Study Procedures

All research procedures and protocols were approved by the Institutional Review Boards at the Universities of Oklahoma and Minnesota. This analysis is a subset of a larger multi-site NIH-funded study MILk (Mothers and Infants Linked for Healthy Growth) in which 365 mother/infant dyads were recruited [30]. The original purpose of the MILk study was to assess the prospective associations of appetite-regulating hormones, growth factors, and inflammatory factors in HM with infant size, adiposity, and lean body mass at 1 and 6 months of age in healthy term infants born to women with a broad range of pre-pregnancy body mass indexes (BMI-18.5–40.0 kg/m^2^). Clinical information was obtained from medical records and questionnaires. Participants were grouped according to maternal pre-pregnancy BMI, with BMI 18.5–25 kg/m^2^ defined as normal-weight (NW) and BMI ≥ 25 kg/m^2^ defined as overweight (OW)/obese (OB). Mother/infant dyads were excluded if they reported tobacco or alcohol use (≥1/week) during pregnancy or lactation, had a history or current presence of diabetes mellitus, or were unable to speak or understand English. Infants were excluded if there was a presumed or known congenital, metabolic, or endocrine disease, or congenital illnesses affecting infant feeding or growth. Mother–infant pairs reported to the study site at 1, 3 and 6 months (±5 days) postpartum, between 08:00 and 10:00, and >1.5 h since the last infant feeding. Upon arrival, a pre-feeding infant weight was obtained using a high-sensitivity scale (Seca 728). Mothers then breastfed their infant ad libitum and as usual from one or both breasts. Following feeding, mothers were assessed for body composition and anthropometrics, and asked to complete questionnaires. After 2–2.5 h post feeding, a complete single-breast milk expression was obtained using a hospital grade breast pump and a standard protocol as described previously [7]. Milk was stored at −80°C within 20 minutes of collection. For the purpose of the current study, 60 mother/infant dyads (30 NW and 30 OW/OB) at the Oklahoma site were randomly selected as a subset from this larger MILk cohort for miRNA analysis. The subjects were matched by maternal age, gestational age at birth, and infant sex. Milk samples collected at 1 and 3 months postpartum and infant growth and body composition measures collected at 1, 3, and 6 months of age were used for analysis.

At 1, 3, and 6 months, infant weight was measured using a Seca 728 scale and length (crown-to-heel) was measured with a Seca 416 infantometer (Seca). The 1- and 3-month infant body composition (% body fat, fat mass [FM], fat free mass [FFM] were measured using the Pea Pod as described previously [31]. The 6-month infant body composition (% body fat, FM, FFM) was measured using dual energy X-ray absorptiometry (DXA; Lunar scanner, GE Healthcare), as previously described [32]. To minimize variability, the same investigator (DAF) positioned the infants and performed all DXA scans.

### 2.2. Exosome Isolation and miRNA Analysis

Samples (2 mL) of HM were thawed on ice and centrifuged twice (1200× *g*, 4 °C, 10 min) to remove fat, cells, and large debris. Defatted supernatant was centrifuged again (21,500× *g*, 4 °C, 50 min) to remove residual fat and casein and subsequently passed through 0.22-µm filters to remove residual cell debris, as previously described by Xi et al. [26]. Exosomes were extracted from 250 µL of defatted filtered milk using ExoQuick^®^ Exosome Precipitation Solution (System Biosciences, Mountain View, CA, USA) according to the manufacturer’s protocol. Total RNA was extracted from the isolated exosomes using the SeraMir Exosome RNA Isolation kit (System Biosciences, Mountain View, CA, USA) according to the manufacturer’s instructions.

A pool of miRNA-specific RT primers was created according to the Applied Biosystems protocol (publication part number 4465407). RT was performed on 20 µL of total RNA using the TaqMan MicroRNA Reverse Transcription Kit (Life Technologies, Grand Island, NY, USA). The full names, TaqMan assay IDs, and sequences of each of the selected miRNAs are available in Appendix A. All quantitative PCR (qPCR) reactions were performed in triplicate using TaqMan MicroRNA Assays and TaqMan Universal Master Mix II, (no UNG) using the Bio-Rad (Hercules, CA, USA) CFX96 Touch Real-Time PCR Detection System. The same starting volume, i.e., 250 µL, of milk was used for RNA extraction for all milk samples. To ensure reproducibility and to check for day-to-day and plate-to-plate variability, a set of the same samples were repeated on each plate (one per plate) as internal controls. The coefficient of variation (CV) for the highly abundant (and thus considered sentinel) miRNAs remained <10% (CV for miR-148a was 9%, miR-30b was 4% and miR-let-7a was 6%). Relative expression by qPCR of individual miRNA species was calculated using the 2−ΔΔCt method [33] in which the geometric mean of all the samples was used for normalization of data [34]. ΔCt was calculated by subtracting the Ct (cycle threshold) value of the experimental sample from the geometric mean of all the samples. The ΔΔCt was calculated by subtracting the ΔCt of the experimental sample (OW/OB group) from the control sample (NW group). The fold change expression of miRNA in the OW/OB group as compared with the NW group was calculated by raising the −ΔΔCt to the power 2. Relative expression levels of miRNA of the OW/OB group compared with the NW group were used in the linear regression models for examining associations between miRNA and infant anthropometrics at 1 and 3 months of lactation. Similarly, fold change expression of miRNA measured at 3 months compared with those measured at 1 month of lactation was calculated. miRNA abundance is the relative abundance or the fold change calculated using NW as a reference group for NW to OW/OB comparison and 1 month as a reference group for 1 to 3 month comparison.

### 2.3. Statistical Analysis

Univariate analyses, including participants’ demographic and anthropometric characteristics, were analyzed using GraphPad Prism statistical software (Version 9.0.0, San Diego, CA, USA) with a standard alpha of 0.05. Normality was determined using the Kolmogorov-Smirnov test. Between-group differences of normally distributed, continuous data were compared using unpaired *t*-test; non-normal data were analyzed by Mann–Whitney U. Linear regression of multiple continuous variables was used to examine the relationship between infant outcome measures (one- and six-month infant weight in kg, % body fat [PBF], fat mass [FM], fat free mass [FFM], and miRNA fold change where 1-, 3- and 6-month infant outcome measures served as dependent variables and 1- and 3-month milk miRNA fold change served as independent variables. The models were controlled for gestational age, birth weight, and infant sex. The three most highly abundant miRNAs (based on the cycle threshold in the qPCR) were selected for analysis at 3 months of lactation. Fold change expression of miRNA collected at 3 months compared with 1 month of lactation was calculated and was used to compare changes in miRNA abundance from 1 to 3 months of lactation. Paired *t*-tests were performed to examine differences in miRNA abundances from the same women at 1 month and 3 months of lactation when differences were normally distributed. Mann–Whitney U tests were performed when differences were not normally distributed. Analysis of covariance (ANCOVA) was performed using relative abundance for each of the three miRNAs to test for differences between 1-month and 3-month miRNA while adjusting for mother’s weight status. 

## 3. Results

### 3.1. Clinical Characteristics of Mother-Infant Dyads

Demographic and clinical information for the mother-infant dyads is presented in Table 1. No significant differences were noted in maternal characteristics or infant anthropometrics at 1, 3, and 6 months of age. At the 1-month post-partum study visit, all the infants were exclusively breastfed; at 3 months, all infants in the NW group and 91% in the OW/OB group were exclusively breastfed; and by 6 months, the rate of exclusive breastfeeding was reduced to ~60%. The percentage of infants that were exclusively breastfed at 6 months of age was not different between NW and OW/OB groups (Table 1.). All 60 mothers were able to provide milk samples at 1 month for analysis. Milk collected from 48/60 mothers was available for analysis at 3 months. For the 12 mothers who did not provide milk samples at 3 months, 2/12 did not exclusively breast feed at 3 months; exclusive breastfeeding status on 5/12 mothers was not known; and 5/12 mothers exclusively breastfed at 3 months of lactation. A total of 48/60 paired 1- and 3-month samples were available for miRNA analysis.

### 3.2. Maternal Overweight/Obesity Decreases Abundance of miRNAs at 1-Month of Lactation

MiRNA fold change in HM collected at 1 month tended to be lower in women with OW/OB than in women with NW. This difference was statistically significant for miR-148a (lower by 30%) and miR-30b (lower by 42%) (Figure 1). There were no significant differences in miRNA fold change between the NW and OW/OB groups in HM obtained at 3 months post-partum (for miR-148a-*p* = 0.99, miR-30b-*p* = 0.71, and miR-let-7a-*p* = 0.85).

### 3.3. Relationship between miRNA Abundance and Infant Anthropometric Measures

(a)MiRNAs sampled at 1-month were associated with infant anthropometrics at 1 month

miR-148a and miR-30b were both significantly associated with infant weight and body composition measures at 1 month, even while controlling for infant gestational age, birth weight, and infant sex (Table 2). For every fold increase in miR-148a, 1-month infant weight, FM, and FFM decreased by 0.61 kg, 0.29 kg, and 0.35 kg, respectively. For every fold increase in miR-32, 1-month infant weight decreased by 0.39 kg. In contrast, the abundance of miR-30b was positively associated with infant anthropometrics. For every fold increase in miR-30b abundance, 1-month infant weight, % body fat, and FM increased by 0.61 kg, 5.62%, and 0.37 kg, respectively.

When the relationships between 1-month HM miRNA abundance and anthropometrics at age 3 months were examined, the positive relationship between miR-30b and infant weight persisted (Table 2). Models examining miRNA measured at 1 month with 6-month anthropometrics and infant body composition in all subjects did not establish any significant association between miRNA fold change and infant anthropometrics. However, when restricted to exclusively breast-fed infants, miR-148a and miR-30b were significantly associated with 6-month infant anthropometrics (Table 2). For example, for every fold increase in miR-148a, 6-month infant % body fat and FM increased by ~5.6% and 0.95 kg, respectively. In contrast, miR-30b showed a negative relationship with 6-month infant FM and FFM.

(b)MiRNAs sampled at 3 months were not associated with infant anthropometrics at 3 and 6 months

Using a similar analytic approach, the relationship between miRNA collected at 3 months of lactation and infant anthropometric measures was examined. There were no associations that passed the threshold for statistical significance for any of the infant outcomes at 3 or 6 months (Table 3).

### 3.4. Levels of miR-148a and miR-30b Differ at 1 and 3 Months Post-Partum

To examine changes of HM miRNA composition between 1 and 3 months post-partum, three miRNAs that were among the most abundant (i.e., those with the lowest absolute Ct values in the qPCR) at the 1-month time point (miR-148a, miR-30b, miR-let-7a) were selected for further analysis (Figure 2). There was a 43% decrease in the fold change of miR-148a (*p* =< 0.0001) and a fourfold increase (*p* =< 0.0001) in the fold change of miR-30b from 1 month to 3 months post lactation when the NW and OW/OB groups were combined (Figure 2A–C). ANCOVA did not show significant interaction between BMI category and miRNA fold change, suggesting that changes in relative abundance of miRNAs from 1 to 3 months of lactation were independent of maternal BMI. There was no difference in the fold change of miR-let-7a (*p* = 0.843) from 1 month to 3 months post lactation. No significant difference was noted between the change in miRNA fold change (3 month-1 month) in NW and OW/OB subjects (miR-148a-*p* = 0.07376, 30b-*p* = 0.07376, let-7a-*p* = 0.5249).

## 4. Discussion

Mounting evidence indicates that milk miRNAs are bioavailable and thus may play a critical role in infant growth and development [11,35,36]. There is also growing evidence suggesting that the composition of HM is dynamic and can be affected by a variety of maternal and infant factors, such as maternal weight, diet, and infant sex [7,9,10,37]. However, limited studies have examined the impact of maternal factors, and specifically maternal obesity, on the abundance of miRNAs in the HM exosomes and its relationship with infant outcomes. The present study provides a comprehensive evaluation of relationships between select HM exosomal miRNAs with maternal overweight/obesity and infant anthropometric measures in the first six months of life. We found that miRNAs-148a and 30b were differentially expressed in HM from OW/OB mothers compared with NW mothers, varied during the course of lactation, and were associated with measures of infant growth and body composition.

The abundance of exosomal miR-148a and miR-30b was reduced in HM obtained from OW/OB mothers compared with HM obtained from NW mothers at one month of lactation. Maternal obesity and gestational diabetes mellitus have been shown to alter miRNA content and expression levels in cord serum exosomes, human umbilical vein endothelial cells, and placenta during perinatal development [38,39,40]. The effects of maternal obesity on HM miRNA content have been examined in two different studies. In line with the findings of the present study, Xi et al. [26] showed that adipogenesis-related miRNAs, including miR-let-7a, miR-30b, and miR-378, were reduced in women with higher BMI [26]. Mirza et al. [41]. found no relationship between maternal BMI and HM exosomal miRNAs in women with type 1 diabetes mellitus as compared with controls without diabetes. However, Mirza et al [41]. analyzed a completely different set of miRNAs (miR-4497, miR-133a, miR-1246, miR-1290, miR-518e-3p, miR-629-3p, and miR-200c-5p) than did Xi et al. and our group, which could explain the differences in the findings. The exact biological significance of the lower abundance of miR-148a and 30b in HM obtained from OW/OB mothers is not known. However, decreased miR-148a might protect infants of OW/OB mothers from becoming overweight. Further, the impact of maternal OW/OB on miRNA did not hold at 3 months of lactation, suggesting HM miRNA function is dynamic over the course of lactation. The first few days of life, specifically the first month of life, is the critical period when relative abundance of select miRNAs, specifically miR-148a and miR-30b is impacted by maternal metabolic status.

The other most striking and novel findings we report are the correlations between the abundance of HM miR-148a (negative) and miR-30b (positive) with anthropometric measures during early life. Higher milk miR-148a was associated with lower infant weight, FM, and FFM at 1 month. Genes targeted by miR-148a affect pathways crucial in adipogenesis, insulin signaling, and energy metabolism [12]. For example, miR-148a targets AMP-activated protein kinase (AMPKα1), a major regulator of energy metabolism, via stimulation of glucose uptake and fatty acid oxidation. miR-148a is shown to directly bind to 3′UTR of insulin-like growth factor receptor (IGF-1R) and inhibit IGF-1R-mediated phosphoinositol 3-kinase (PI3K)/AKT activation, suggesting that miR-148a may affect cell growth via IGF1R regulation [42]. The negative relationship between HM miR-148a and infant weight may be supported by 148a mediated inhibition of IGF-1R [43].

Higher miR-30b was associated with higher infant weight, % body fat, and fat mass at 1-month of age. While little is known about the role of miR-30b in growth and development, emerging data point to a role of the miR-30 family in promoting adipogenesis in both humans and mouse models [44]. Brown adipose tissue and inducible brown adipocytes (beige or BRITE) are rich in mitochondria and dissipate energy in the form of heat. MiR30b/c overexpression induces mitochondrial and thermogenic genes (*Ucp1*, *Cidea*) and mitochondrial respiration in brown and BRITE adipose tissue in mouse models [45]. Further, some studies suggest that miR-30 species may play a role in white fat adipogenesis [46,47]. For example, overexpression of miR-30b can stimulate adipogenesis, increase the size of lipid droplets, and upregulate lipogenic genes [47]. Whether HM miR-30b affects adipogenesis or browning in nursing infants remains to be determined.

To date, only one other report has examined the relationship between HM miRNAs and infant growth. Assessing a completely different set of miRNAs, Zamanillo et al. [27]. measured miRNAs in whole breast milk that target mRNA of leptin (LEP), adiponectin (ADIPOQ) and their respective receptors (LEPR, ADIPOR1 and ADIPOR2) which are associated with obesity. They reported that the abundance of HM miR-103, miR-17, miR-181a, miR-222, miR-let7c, and miR-146b obtained at two months post-partum was negatively correlated with infant BMI at 24 months of age in normal weight, but not obese, mothers [27].

The relationships between the HM miRNAs and infant growth that we report were primarily observed during the first month of life. The exact reasons or underlying mechanisms for this are unknown. However, a few speculations can be made. We know that gestational age, postnatal age, and type of enteral nutrition (breast milk versus formula) play a role in maturation of the gut, resulting in a decrease in intestinal permeability with postnatal age [48,49]. The first month of life represents adaptation of the growing infant to optimize nutrient intake and may be regulated by changes in microbiome from one to six months of life [50]. In addition, our data show overall lower relative abundance of select miRNAs from one to three months of lactation, suggesting that a lower dose of miRNAs might be delivered to the infants at three months than at one month of lactation. Thus, our data support the hypothesis that early life is a period of nutritional programming for the developing infant and a window of opportunity to improve long-term health of infants [51]. Further, exosomes obtained from bovine milk are absorbed systemically and are accumulated in intestinal mucosa, spleen, liver, and brain in the mouse model [35]. Thus, the gastrointestinal tract of the newborn may render the infant uniquely susceptible to the effects of exosome-borne miRNAs in HM, potentially affecting the intestinal epithelium, function or constitution of gut microbiota, or allowing absorption of the miRNA leading to systemic effects. The absence of associations at six months in the group as a whole may be due to the introduction of infant formula, which dilutes the HM miRNA exposure. It is curious, however, that a few relationships appeared to reverse when the one-month HM sample was related to six-month anthropometrics in the exclusively breastfed infants. Though we do not have a ready explanation for this, we think that exosomal miRNAs in HM are one of the protective mechanisms by which breast milk might protect infants born to OW/OB mothers from being OW/OB later in life. For instance, from birth to one month of age, NW infants gain 1.08 kg on average, while infants born to mothers with OW/OB gain 0.81 kg; this delta seems notable. When plotted on WHO weight percentiles, NW infants stay at the 50th percentile, while infants in OW/OB group go from around 75th percentile to the 50th percentile by one month of age. Further, the reversal of some relationships from one to six months of lactation in exclusively breastfed infants might be explained from the dynamic changes that occur in the infant gut functionality, intestinal microbiome, and HM miRNAs that are absorbed systemically in the growing infant to have a functional role.

We also observed changes in the abundance of select miRNAs in HM during the lactation period, indicating that the HM miRNA content is a dynamic feature. Specifically, the abundance of miR-148a declined between one and three months post-partum, whereas the abundance of miR-30b increased over the same period. These observations were consistent in both the NW and OB/OW groups. These findings align with previous studies showing changes in HM miRNA abundance collected during different periods of lactation [23,26,52]. Xi et al. [26] reported that the level of miR-30b is higher in mature milk than in colostrum.

However, none of those studies are directly comparable to ours because HM was sampled at different times, [23,52] miRNA source differed [23] (e.g., cell associated vs exosomal), or the specific miRNAs quantified did not overlap with ours [23,52]. For example, in the study reported by Alsaweed et al., total miRNA concentration in the cell and lipid fraction of the HM did not change in the first six months of lactation [23]. Though miRNAs are abundant in different fractions of breast milk, exosomal miRNAs are of particular interest as they are likely to be bioactive and taken up by the infant gut with some functional role [35]. It is also important to note that in the study described by Alsaweed et al., miRNAs were examined in 10 lactating mothers by deep sequencing versus targeted analysis. In the present study, we related miRNA quantity to milk volume in order to calculate miRNA fold change/abundance and thereby estimate exposure to the infant.

Here, our approach was to focus on a set of highly abundant HM exosomal miRNAs with the potential to influence metabolic processes. Our observations suggest that exosome-borne miRNAs act directly on the nursing infant to affect metabolism and growth. This hypothesis assumes that miRNAs survive in the infant’s gastrointestinal tract and exert effects on intestinal epithelium and/or are absorbed and enter the circulation. Current evidence strongly supports these assumptions [11,20,35,53,54]. Thus, the relatively immature gastrointestinal tract of newborns may allow effective molecular communication between mother and infant through exosomal miRNAs. We speculate that an increase in miR-148a exposure following ingestion could reduce insulin action and thereby slow infant growth and fat acquisition whereas exposure to the increased miR-30b might support the increases in fat mass in the offspring as we observed.

Breastfeeding is believed to affect metabolic disease risk later in life [6,55,56,57] and protect against overweight and obesity in childhood [58,59]. Because infant formula lacks exosomes and viable cells, it contains a much lower concentration of miRNAs [11]. While the exact mechanisms underlying the protective effect of breastfeeding on overweight and obesity remain unknown, we postulate that exosomal miRNAs in HM, along with other bioactives, affect energy balance in the offspring. An extensive review carried out by Melnik at al. describes suppression of DNA methyltransferase (DNMT) by HM miRNAs, specifically miR-148a, which results in hypomethylation and, consequently, activation of a variety of genes affecting growth and development [12]. Melnik also links milk-derived miR-148a to pancreatic beta-cell differentiation and thus suggests a potential mechanism by which breast milk protects against the development of type 2 diabetes mellitus [60].

We acknowledge the limitations of our study. First, the analysis was restricted to six miRNAs in HM. Though we chose to analyze highly abundant miRNAs that are involved in important metabolic pathways, this approach might miss other abundant miRNAs in HM exosomes that might have an important role with respect to infant outcome measures. Second, we used two different methods to measure body composition, Pea Pod at 1 and 3 months and DXA at six months of age. However, both of these methods are standard and validated to measure infant body composition in early infancy [31,32,61,62]. Third, we measured miRNAs in exosomes isolated from skimmed milk and thus could not assess the contribution of lipid or cell-associated miRNAs. However, out of all the fractions of HM, we think that miRNAs packaged within the HM exosomes are most bioavailable to the growing infant. Fourth, there could be a dose effect with miRNAs that are ingested by the infant through breast milk. We do not have infant blood to evaluate miRNAs in systemic circulation of the infant. Future studies will be planned to address this question. In addition, we measured the abundance of miRNA in a single HM sample at each time. The extent to which this single sample represents the average output from a given mother is unknown. Despite this, we were able to find clear relationships between HM miRNA content and indices of infant growth and body composition. Lastly, as this is a correlational study, we cannot establish causal relationships.

Our study has several strengths: First, detailed, longitudinal measures of infant growth and body composition have not been done in any other study relating HM miRNAs effects on infants. Second, we used controlled and consistent sampling protocols for HM acquisition. Third, we carefully selected comparison groups for OW/OB with elimination of some confounders.

## 5. Conclusions

In conclusion, we demonstrated that miRNA-148a and miRNA-30b, both of which are involved in important metabolic pathways are abundant in HM exosomes and that the abundance of these miRNAs is affected by maternal overweight/obesity and lactation duration. We found that miRNAs 148a and 30b are significant predictors of infant growth and fat acquisition in early infancy. Further studies investigating the role of other maternal factors, such as diet and exercise, on miRNA composition in HM are warranted. Further mechanistic studies are needed to identify the biological role of exosomal miRNAs with an eye towards the infant gut, liver and developing β-cells of the pancreas. Changes in the abundance of exosomal miRNAs in HM obtained from the OW/OB mothers might be one of the mechanisms by which breast milk protects the infants from obesity and type 2 diabetes mellitus later in life.

## Figures and Tables

**Figure 1 nutrients-13-01091-f001:**
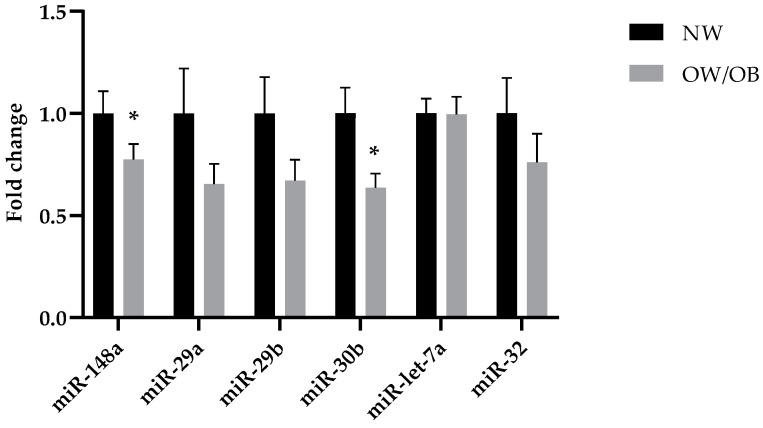
Levels of miRNA-148a and 30b are reduced in milk of overweight (OW)/obese (OB) mothers at 1 month of lactation mean ± Standard Error of Mean (*SEM*). * = *p* < 0.05. Fold change of miR-148a (*p* = 0.04) and miR-30b (*p* = 0.007) was lower in OW/OB mothers than in normal-weight (NW) mothers.

**Figure 2 nutrients-13-01091-f002:**
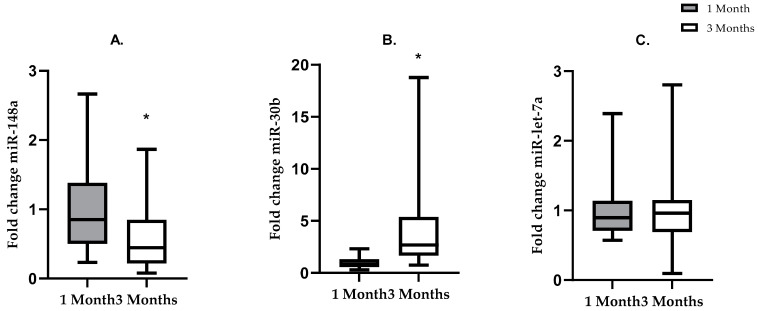
Changes in fold change of miRNAs from 1 to 3 months of lactation in all (both NW and OW/OB) mothers, without adjusting for maternal BMI. Grey bars show miRNA fold change at 1 month of lactation. White bars show miRNA fold change at 3 months of lactation. (**A**) Mean ± SD. * = *p* < 0.05: fold change expression of miR-148a was lower at 3 months than at 1 month of lactation (*t*-test *p* =< 0.0001). (**B**) Mean ± SD. * = *p* < 0.05. Fold change of miR-30b was higher at 3 months than at 1 month of lactation (Mann–Whitney *p* = < 0.0001). (**C**) Mean ± SD. No difference in the fold change of miR-let-7a from 1 to 3 months of lactation (Mann–Whitney *p* = 0.842).

**Table 1 nutrients-13-01091-t001:** Maternal and Infant Characteristics ^1^.

	NW (*N* = 30)	OW/Obese (*N* = 30)	*p* Value ^2^
**Mothers**			
Age	29.9 ± 4	29.5 ±5	0.691
Race, % white	87	83	0.553
Pre-pregnancy BMI, kg/m^2^	22 ± 1.8	31.6 ± 7.3	<0.0001
Gestational weight gain, kg	12.9 ± 5.6	10.6 ± 8.7	0.219
Weight loss 1 to 3 months, kg	1.4 ± 1.6	0.6 ± 3.6	0.318
Weight loss 3 to 6 months, kg	1.7 ± 1.9	1.2 ± 4.8	0.621
Weight loss 1 to 6 months, kg	3.7 ± 3.7	1.8 ± 6.0	0.194
**Infants**			
Sex, % males	53	53	1.000
Birth weight, kg	3.53 ± 0.4	3.74 ± 0.5	0.094
Gestational age (weeks)	39.26 ± 0.4	38.64 ± 0.9	0.887
Exclusively breast fed at 3months, %	100	90.91	0.003
Exclusively breast fed at 6months, %	63.33	60	0.240
**1-month infant anthropometrics and body composition ^3^**			
Weight, kg	4.62 ± 0.64	4.56 ± 0.77	0.754
% Body fat	17.17 ± 6.13	17.55 ± 5.40	0.804
Fat mass, kg	0.80 ± 0.34	0.82 ± 0.33	0.822
Fat free mass, kg	3.80 ± 0.47	3.77 ± 0.61	0.819
**3-month infant anthropometrics and body composition ^3^**			
Weight, kg	6.10 ± 0.63	6.21 ± 1.10	0.635
% Body fat	23.6 ± 5.20	22.70 ± 6.20	0.607
Fat mass, kg	1.50 ± 0.40	1.50 ± 0.60	0.881
Fat free mass, kg	4.60 ± 0.40	4.80 ± 0.70	0.320
**6-month infant anthropometrics and body composition ^4^**			
Weight, kg	7.58 ± 0.66	8.08 ± 1.35	0.099
% Body fat	33.24 ± 3.22	34.89 ± 3.30	0.097
Fat mass, kg	2.58 ± 0.375	2.81 ± 0.77	0.196
Fat free mass, kg	5.00 ± 0.84	5.22 ± 0.692	0.368

^1^ Continuous data are presented as means ± SDs (*n*); categorical variables are presented as a percentage (*n*). ^2^
*p* value calculated using Student’s 2-sided *t*-test for continuous data; Fisher’s exact test used for categorical data. ^3^ 1- and 3-month body composition assessed using air displacement plethysmography. ^4^ 6-month body composition assessed using dual energy X-ray absorptiometry.

**Table 2 nutrients-13-01091-t002:** Associations between 1-month breast milk exosomal miRNA abundance and infant outcomes at 1, 3 and 6 months of age.

	miR-148a	miR-29a	miR-29b	miR-30b	miR-let-7a	miR-32
**1-month outcomes ^1^** **(*N* = 60, all infants)**						
Weight	−0.616 *	0.327	−0.008	0.616 **	−0.027	−0.386 *
% Body Fat	−4.807	0.827	−1.795	5.624 *	−0.890	−0.580
Fat Mass (kg)	−0.291 *	0.089	−0.087	0.374 *	−0.044	−0.093
Fat Free Mass (kg)	−0.352 *	0.242	0.059	0.257	0.004	−0.274
**3-month outcomes ^1^** **(*N* = 60, all infants)**						
Weight	−0.175	0.108	−0.182	0.640 *	−0.511	−0.120
% Body Fat	−1.140	0.277	−3.691	3.023	−1.826	2.228
Fat Mass (kg)	−0.081	0.040	−0.248	0.275	−0.221	0.119
Fat Free Mass (kg)	−0.184	0.046	0.241	0.335	−0.294	−0.290
**6-month outcomes ^2^** **(*N* = 60, all infants)**						
Weight	0.209	0.475	−0.423	−0.256	−0.165	−0.118
% Body Fat	2.938	0.218	−1.462	−0.922	0.307	−0.219
Fat Mass (kg)	0.415	0.305	−0.224	−0.265	0.016	−0.306
Fat Free Mass (kg)	0.494	0.466	−0.036	−0.572	0.375	−0.385
Weight Gain 0–6 Months	−0.051	0.294	−0.535	0.385	−0.314	0.039
**6-month outcomes ^2^** **(*N* = 37, exclusive breastfeeding infants only)**						
Weight	0.550	0.544	−0.865	−0.480	−0.372	0.044
% body fat	5.576 *	−1.268	−0.711	−2.950	0.296	0.884
FM	0.953 *	−0.337	0.137	−0.785 *	0.050	0.262
FFM	1.250	−0.053	0.996	−1.463 *	0.676	−0.413
Weight Gain 0–6 Months	0.816	−0.154	−0.081	−0.474	−0.506	0.304

Each row shows respective β (estimate) for the regression model where infant outcomes are dependent variables and the miRNA fold change are independent variables. ^1^ 1- and 3-month body composition assessed using air displacement plethysmography. ^2^ 6-month body composition assessed using dual energy X-ray absorptiometry * = *p* < 0.05, ** = *p* <0.01.

**Table 3 nutrients-13-01091-t003:** Associations between 3-month breast milk exosomal miRNA abundance and infant outcomes at 3 and 6 months of age in infants still being exclusively breastfed.

	miR-148a	miR-30b	miR-let-7a
**3-month outcomes ^1^ (*N* = 48, all infants)**			
Weight	−0.191	0.136	−0.083
% Body Fat	−1.76	2.310	−1.307
Fat Mass (kg)	−0.130	0.123	−0.062
Fat Free Mass (kg)	−0.161	0.041	−0.057
**6-month outcomes ^2^ (*N* = 48, all infants)**			
Weight	−0.227	−0.087	0.043
% Body Fat	0.062	−0.276	0.406
Fat Mass (kg)	−0.322	0.170	−0.186
Fat Free Mass (kg)	0.165	−0.100	0.040

Each row shows respective β (estimate) for the regression model where infant outcomes are dependent variables and the miRNA fold change are independent variables. ^1^ The 3-month body composition assessed using air displacement plethysmography. ^2^ The six-month body composition assessed using dual energy X-ray absorptiometry (DXA).

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
