# Peer review of "Human Milk Exosomal MicroRNA: Associations with Maternal Overweight/Obesity and Infant Body Composition at 1 Month of Life"

_nutrients, 2021, doi:10.3390/nu13041091_

Round 1

Reviewer 1 Report

This well written manuscript interestingly evaluates the role of maternal weight on longitudinal human milk microRNA composition and the impact on infant body composition over the first 6 months of life. This topic is of high importance as there is limited knowledge on the dynamic composition of microRNAs in human milk and the role of maternal factors influencing composition. This manuscript provides support to the idea that human milk microRNAs may have long term impacts on infant growth; however, evaluates only a small targeted set of microRNAs. Further clarity on statistical analysis methods and interpretation of the growth results would provide further strength to this manuscript.

The following are specific concerns to be addressed to improve the clarity of the manuscript:

Abstract:

  • Include the number of milk samples available for 3 month analysis as the n is limited further (48/60)

Introduction

  • Consider discussion of what is currently know in the limited literature evaluating milk microRNAs and maternal influences (Zamanillo 2019, Xi 2015) & how your work with contribute to this gap in knowledge

Materials and Methods

  • Pg 3, Ln 127 – Please clarify “The miRNA abundance was related to volume of breast milk”. Was this adjusted for when comparing milk samples of miRNA expression?
  • Pg 3, Ln 131 – Was the normalized microRNA expression vs the relative expression (compared to NW or 1m time frame) used for further analysis, this is not clear particularly in relation to the growth associations.
  • Pg 3, Ln 146 – Clarify what is meant by microRNA abundance vs fold change throughout the manuscript.
  • Pg 4, Ln 155 – Clarify the microRNA expression used for ANCOVA analysis, was this overall abundance or a relative expression?
  • Did you consider evaluating broader set of microRNAs in human milk given the limited evaluation in the current literature or including those previously evaluate microRNA associated with infant growth?

Results

  • 1: Were all 60 mothers able to provide milk samples at 1 month to include in analysis?

Were the 12 mothers that did not provide milk samples at 3 months exclusively formula feeding?

  • How many paired 1m and 3m milk samples were available?
  • Figure 1. Consider showing only the fold change of OW/OB relative to NW based on delta delta CT as you described rather than side by side bars
  • 3 Clarify the definition of miRNA abundance
  • 4: Where the most abundant based on normalized Ct values or raw Ct values?
  • Figure 2. Was maternal BMI adjusted for?

Discussion

  • The limited early impact of milk microRNA on infant growth without long term impacts is interesting, although as stated well in the manuscript requires further investigation including investigation longer term into metabolic disease risk. Could there be a dose effect with microRNA exposure through breast milk? Or do you have infant blood to evaluate systemic microRNA levels?

Reviewer 2 Report

The authors examine associations of maternal overweight/obesity with select miRNAs in human milk and, in turn, associations between these miRNAs and measures of infant weight and body composition across the first 6 months of life. This paper is of interest to the field of maternal-child health and pediatric obesity. The manuscript is well-written and strengthened by the highly standardized breast milk collection procedure and assessment of infant adiposity. The authors do a good job clearly outlining the limitations of the study. Although the authors acknowledge having no ready explanation for why the relationships between miR-148a and miR-30b reverse from 1 to 6-months of infancy, it would strengthen the manuscript to include proposed hypotheses for why this may occur or the potential significance of this reversal in the direction of the association. Additionally, I am curious as to why the authors chose to adjust models predicting infant fat mass for birth weight rather than for fat-free mass at the assessment timepoint or (for 3-month and 6-month assessments) fat mass at 1-month. Otherwise, I have no additional suggested revisions.

Reviewer 3 Report

This study examined the impacts of maternal overweight/obesity on select miRNAs (miR-20 148a, miR-30b, miR-29a, miR-29b, miR-let-7a and miR-32) and relationship of these miRNAs with infant body composition in the first 6 months of life. This analysis is a subset of a larger multi-site NIH funded study MILk. The analyses appear to have been conducted appropriately, and the manuscript is well written. Below are specific comments that should be addressed.

  1. It is known that to carry out the DXA it is necessary that the individual be immobile. Please explain how you managed to minimize movement artifacts during DXA scanning.

  1. Since two different methods were used to measure body composition, Pea Pod at 1 and 3 months and DXA at 6 months of age. Please provide the precision of the Pea Pod and DEXA methods used to determine tissue mass.

  1. Line 177: “There were no significant differences in miRNA fold change between maternal weight status groups in HM obtained at 3-months post-partum”. Please justify why that may happen? Since the composition of HM is dynamic, it might be related to maternal weight changes at 3 and 6 months? Maternal weight loss (or changes) during the course of lactation (3 and 6 months) was not included in the Table 1. Please include this information. 

  1. As you mentioned the composition of HM is dynamic and can be affected by a variety of maternal and infant factors. Have you assessed the impacts of maternal dietary intake during pregnancy and/or lactation on the HM content of select miRNAs?

  1. What would be possible mechanisms/explanations for the lack of associations between the HM miRNA and infant growth at 3 and 6 months?
